# Multidimensional Frailty Predicts Mortality Better than Physical Frailty in Community-Dwelling Older People: A Five-Year Longitudinal Cohort Study

**DOI:** 10.3390/ijerph182312435

**Published:** 2021-11-26

**Authors:** Alberto Cella, Nicola Veronese, Monica Pomata, Katerin Leslie Quispe Guerrero, Clarissa Musacchio, Barbara Senesi, Camilla Prete, Erica Tavella, Ekaterini Zigoura, Giacomo Siri, Alberto Pilotto

**Affiliations:** 1Department of Geriatric Care, Orthogeriatrics and Rehabilitation, Galliera Hospital, 16128 Genova, Italy; alberto.cella@galliera.it (A.C.); monica.pomata@galliera.it (M.P.); katerin.quispe@galliera.it (K.L.Q.G.); clarissa.musacchio@galliera.it (C.M.); barbara.senesi@galliera.it (B.S.); camilla.prete@galliera.it (C.P.); erica.tavella@galliera.it (E.T.); ekaterini.zigoura@galliera.it (E.Z.); alberto.pilotto@galliera.it (A.P.); 2Geriatric Unit, Department of Medicine, University of Palermo, 90133 Palermo, Italy; 3Scientific Directorate Biostatistics, Galliera Hospital, 16128 Genova, Italy; Giacomo.siri@galliera.it; 4Department of Interdisciplinary Medicine, University of Bari Aldo Moro, 70121 Bari, Italy

**Keywords:** frailty, multidimensional prognostic index, prognosis, mortality

## Abstract

Frailty is a common syndrome in older people that carries an increased risk of mortality. Two main models describe frailty, either as a loss of physical functions or as an accumulation of multiple deficits. The aim of our study was to compare the physical frailty index developed in the Cardiovascular Health Study (CHS) with a multidimensional frailty tool, the Multidimensional Prognostic Index (MPI), in predicting death in community-dwelling older subjects. Four hundred and seven community-dwelling older subjects were enrolled. Each subject underwent a comprehensive geriatric assessment (CGA) with calculation of the MPI and CHS index. Mortality was recorded over the following 5 years. In the overall sample (mean age of 77.9 ± 4.5 years; 51.6% female), 53 subjects (13%) died during the 5-year follow-up period. Both the MPI and CHS index were able to predict mortality; however, the MPI was significantly more accurate than the CHS index in predicting mortality (C-index = 0.69 and 0.59, respectively; *p* < 0.001), with a statistically significant difference of 10%. In conclusion, multidimensional frailty, assessed by the MPI, predicts five-year mortality in community-dwelling older people better than physical frailty, as assessed by the CHS index. These findings suggest the usefulness of assessing frailty by means of CGA-based tools to predict relevant health-negative outcomes in older people.

## 1. Introduction

Frailty is a condition characterized by a decline in functioning across multiple physiological systems, accompanied by elevated vulnerability to stressors [1]. It is a common condition in older people, with some data indicating that one person in ten suffers with this condition in any given community [2]. This condition is associated with several negative outcomes in older people, including cardiovascular disease [3], mood disorders [4], disability [5], hospitalization [6], and, ultimately, mortality [7].

Frailty is commonly defined in geriatric medicine by a variety of models. One of the most widely known is the phenotypic model, which identifies this specific condition by the presence of three or more of the following five features: (a) unintentional weight loss; (b) exhaustion; (c) low physical activity level; (d) slow walking speed; (e) muscle weakness [8]. Another model is the multiple-deficits model, which rates frailty through the number of functional, sensory and clinical deficits [9]. These two models capture different trajectories of frailty in older people. However, studies based on the two different definitions produced quite different results [2]. More recently, a new concept of frailty has emerged, i.e., the multidimensional model [10,11]. According to this model, frailty is the loss of harmonic interaction across multiple domains or dimensions, ultimately leading to homeostatic instability. This is better captured by means of the comprehensive geriatric assessment (CGA) [10]. It was largely reported that the multidimensional aspects of this condition, captured by the CGA, can be reported using the Multidimensional Prognostic Index (MPI) [10,12]. A large amount of literature has shown that the MPI is a well-calibrated and accurate tool, highly predictive of negative outcomes in older people, as reported in more than 55,000 individuals [10,12].

Unfortunately, no study, to the best of our knowledge, has compared the long-term predictive value of the MPI versus the model proposed in the Cardiovascular Health Study (CHS) in community-dwelling older people [13]. With this research, we aimed to compare the predictive power of the MPI vs. the CHS tool in predicting death in community-dwelling older subjects over five years of follow-up.

## 2. Materials and Methods

### 2.1. Participants

Approximately 1500 subjects aged 65 years and over, equally distributed by gender and age class, were randomly extracted and contacted by means of recruitment notices, mails and phone calls over three districts of Genoa, Italy.

Preliminary inclusion criterion was the availability to complete a 30-min face-to-face questionnaire (lifestyle and socio-economic factors) and to undergo a medical examination and a CGA in our outpatient clinic. Overall, the response rate was 27.2% of all subjects contacted; therefore, 407 community-dwelling older subjects were enrolled starting from 14th April 2014.

The local ethics committee approved the study protocol, and participants gave their written informed consent to the study.

### 2.2. Exposure: Multidimensional Prognostic Index and Cardiovascular Health Study Criteria

Each subject underwent a CGA with calculation of the MPI and the CHS frailty phenotype.

CHS criteria defined frailty by using 5 measurable items (unintentional weight loss, low physical activity level, weakness, exhaustion, and slow gait speed) [8]. In the present study, we used the same measurement tools as those used in the CHS study, apart from the Physical Activity Scale for the Elderly (PASE) for measuring usual physical activity, as in a previous study on the same sample [14], which are as follows:

1. Weight loss: unintentional weight loss of more than 5% of body weight in the previous year;

2. Weakness: grip strength in the lowest 20% at baseline, adjusted for gender and body mass index (BMI);

3. Poor endurance and energy: self-reported exhaustion, identified by the same criterion used in the original description of the ‘frail’ phenotype (two questions from the CES–D scale);

4. Slowness: low gait speed in a 4 m walk at usual pace, normalized by gender and height (i.e., men: height ≤173 cm and speed ≤0.65 m/s; height >173 cm and speed ≤0.76 m/s; women: height ≤159 cm and speed ≤0.65 m/s; height >159 cm and speed ≤0.76 m/s);

5. Low physical activity level: PASE score [15] in the lowest quintile of the study sample, adjusted for gender.

Participants were classified as follows: (a) frail if they met 3 or more of the 5 Fried criteria; (b) pre-frail if they met 1 or 2 criteria; (c) not frail if they met none of the criteria.

The MPI was calculated from information obtained through a standard CGA that considered the following eight different domains [16,17]:

1. Functional status as evaluated by Katz’s Activities of Daily Living (ADL) index [18], which defines the level of dependence/independence in six daily personal care activities (bathing, toileting, feeding, dressing, urine and bowel continence and transferring (in and out of bed or chair)).

2. Independence in the Lawton’s Instrumental Activities of Daily Living (IADL) [19], which assesses independence in eight activities that are more cognitively and physically demanding than ADL, i.e., managing finances, using a telephone, taking medications, hopping, using transportation, preparing meals, doing housework and washing.

3. Cognitive status, which is measured through the Short Portable Mental Status Questionnaire (SPMSQ) [20], a ten-item questionnaire investigating orientation, memory, attention, calculation, and language; validated versions were used in each local language.

4. Co-morbidity was examined using the Cumulative Illness Rating Scale (CIRS) [21]. The CIRS uses a 5-point ordinal scale (score 1–5) to estimate the severity of pathology in 13 different systems, including cardiac, vascular, respiratory, eye–ear–nose–throat, upper and lower gastrointestinal, hepatic, renal, genitourinary, musculoskeletal, skin disorders, nervous system, endocrine–metabolic and psychiatric behavioral disorders. Based on the ratings, the Comorbidity Index (CIRS-CI) score, which reflects the number of concomitant diseases, was derived from the total number of categories in which moderate or severe levels (grade from 3 to 5) of disease were identified (range from 0 to 13).

5. Nutritional status was investigated with the Mini Nutritional Assessment (MNA) [22], an 18-item questionnaire comprising anthropometric measurements (BMI, mid-arm and calf circumference, and weight loss) combined with a questionnaire regarding dietary intake (number of meals consumed, food and fluid intake, and feeding autonomy), a global assessment (lifestyle, medication, mobility, presence of acute stress, and presence of dementia or depression), and a self-assessment (self-perception of health and nutrition).

6. Risk of developing pressure sores was evaluated through the Exton-Smith Scale (ESS), a five-item questionnaire determining physical and mental condition, activity, mobility and incontinence [23].

7. Number of medications taken daily was categorized as <3, 4–7, >7, and did not consider topical medications.

8. Cohabitation status included living alone, in an institution, or with family members.

For each domain, a tripartite hierarchy was used, i.e., 0 = no problems, 0.5 = minor problems, and 1 = major problems, based on conventional cut-off points derived from the literature for each item. The sum of the calculated scores from the eight domains was divided by 8 to obtain a final MPI risk score ranging from 0 = no risk to 1 = higher risk of mortality [16]. Traditionally, the categorization of MPI is made using three categories, i.e., MPI-1 (low risk of mortality) <0.33; MPI-2 (moderate risk) between 0.33 and 0.66; MPI-3 (high risk) with an MPI value >0.66 [16].

The MPI assessment requires between 15 and 25 minutes, and the results can be automatically generated using the MPI calculator software 1.2.0 downloaded from the https://multiplat-age.it/index.php/en/tools (accessed on 25 November 2021).

### 2.3. Outcome: Mortality

All-cause mortality was recorded, using administrative data, over the following 5 years, until 1 June 2020.

### 2.4. Statistical Analysis

Data were summarized by means of the common descriptive statistics (mean, standard deviation and range for quantitative variables, absolute and relative frequencies for qualitative variables). The agreement between MPI and CHS index was measured using the Kappa coefficient. Cox proportional hazards models were used to assess the effect of CHS and MPI on mortality, adjusting for confounders. The discriminative capability of the two indexes was estimated in terms of C-index (95% CI) and their difference. We also reported data on the discriminative capability of a model, including age and sex. The statistical significance was set to 5%. The analysis was conducted with STATA software (version 14.1, StataCorp LP, College Station, TX, USA).

## 3. Results

Table 1 shows the baseline descriptive data of the study population. In the sample as a whole (mean age of 77.9 ± 4.5 years; 51.6% female), at the baseline, the prevalence of physical frailty, according to the CHS index, was 9.3%, and of pre-frailty, it was 26.5%. According to the MPI score, 2% of the subjects were in the high-risk category (MPI-3) and 18% were in the moderate-risk category (MPI-2). Using these data, all the people included in the MPI-3 group were classified as frail using the CHS index, but only 30/108 of the subjects included in the MPI-2 group were ranked as pre-frail. Therefore, the agreement between the MPI and the CHS index was poor (kappa = 0.21; *p* < 0.0001). Dividing the participants according to survival status, people who died during the follow-up period were significantly older, had higher presence of frailty, according to the MPI (*p* < 0.0001) and the CHS index (*p* = 0.002), and scored significantly worse in several domains of the MPI and the CHS index (Table 1).

During the 5-year follow-up period, 53 subjects (13%) died. According to the CHS index, taking the robust class as a reference, and after adjusting for age and sex, frailty was associated with a significantly higher risk of mortality (HR = 2.63; 95%CI: 1.28–5.36; *p* = 0.008), whilst pre-frailty was not predictive of mortality (HR = 1.21; 95%CI: 0.63–2.31; *p* = 0.57). A similar analysis was run for the MPI, the results of which are as follows: taking the participants in MPI-1 category as a reference (not frail), the MPI-2 category (moderate multidimensional impairment) was associated with a significantly higher risk of death (HR = 4.16; 95%CI: 2.30–7.52; *p* < 0.0001), as was MPI-3 (HR = 23.17; 95%CI: 7.16–75.03; *p* < 0.0001). As shown in Table 2, both the MPI and the CHS index were able to predict mortality. However, the MPI was significantly more accurate than the CHS index in predicting mortality (C-index = 0.69 and 0.59, respectively; *p* < 0.001), with a statistically significant difference of 10% (95%CI 0.02–0.18, *p* = 0.013) (Table 2). Importantly, age and sex together did not significantly predict death during the follow-up period (C-index = 66.6; 95%CI: 58.9–74.4; *p* = 0.188).

## 4. Discussion

In this study, we compared, for the first time, the ability to predict mortality of multidimensional frailty, as assessed by the MPI, and physical frailty, as assessed by the CHS index. Overall, our results indicated that multidimensional frailty is more strongly associated with mortality than physical frailty, suggesting that the CGA, beyond physical evaluation, is needed to identify different risks of mortality in older people.

In geriatric medicine, it is widely known that there is no gold standard for measuring frailty. Therefore, many different frailty instruments have been created and proposed, following different conceptual models [24]. Unfortunately, the agreement by which individuals can be classified as frail, according to different tools, is poor [25]. In this sense, our study also showed that the agreement between the CGA-based MPI and the CHS index is poor. To fully assess and compare the performance, accuracy, and validity of different frailty tools, it is also important to consider their prospective association and predictive ability regarding the main conditions associated with frailty and, in particular, mortality [26]. In this sense, a pivotal paper, including 35 different tools for evaluating frailty and mortality, in the English Longitudinal Study on Ageing (ELSA), which followed more than 5000 older participants for 7 years, found that multidimensional frailty scores may have a stronger and more stable association with mortality than tools assessing physical frailty [27]. However, to the best of our knowledge, our study is the first to compare the ability of the MPI to predict mortality compared to physical frailty in community-dwelling subjects, further supporting the use of the MPI, a tool initially validated in hospitalized older people, in primary care settings as well.

According to the multidimensional model, a novel model for approaching frailty, the identification of this condition could be approximated by the CGA, particularly if a tool may convey information in biological, functional, psychological, clinical, and social domains [28]. Increasing literature, carried out in different clinical settings and on several specific diseases, has shown that routine application of the CGA significantly reduces mortality in older people [10]. Moreover, the CGA has a positive influence not only on mortality, but also on other outcomes, such as institutionalization, hospitalization, and the functional and cognitive status of older patients [10]. The CGA, in fact, might help ensure good appropriateness of prescribing and intervention in frail older adults, therefore reducing all the previously mentioned negative outcomes [29]. Importantly, a recent paper on frailty, by the European Medicines Agency, reported that a complete evaluation of frailty, to support its management, requires a multidimensional interdisciplinary CGA [30]. In this context, the MPI is recognized as a CGA-based predictive tool, able to extract information from a standard CGA with an excellent prognostic value [30]. Indeed, our actual work supported the predictive power of MPI compared to physical frailty tools.

The findings of our study should be interpreted with the following limitations taken into consideration: the cohort included a relatively limited sample size and only eight people were included in the MPI-3 category, potentially limiting our results. In this regard, in fact, the study population was mainly composed of fit older subjects (80.1% in the MPI-1 category and 64% in the robust category of CHS), with an oversampling of this population, being poorly generalizable in other contexts in which multidimensional and physical frailty have a high prevalence. Finally, physical frailty was measured in compliance with the Cardiovascular Health Study Criteria, but not using the same criteria as those used in the Cardiovascular Health Study. Therefore, as widely known, this can decrease the predictive value of physical frailty in predicting mortality [31].

## 5. Conclusions

Multidimensional frailty, assessed by the CGA-based MPI, predicts five-year mortality in community-dwelling older people better than physical frailty, as assessed by the CHS index. Other longitudinal studies are needed to confirm the importance of assessing multidimensional frailty, to better predict outcomes other than mortality.

## Figures and Tables

**Table 1 ijerph-18-12435-t001:** Descriptive data of the overall sample.

Item	Mean (SD) or Number (%)	Range	Death = No 354 (87.0%)	Death = Yes 53 (13.0%)	*p*-Value
Female gender	210 (51.6)	-	185 (52.3)	25 (47.2)	0.556
Age (years)	77.9 (4.5)	(65–90)	77.5 (4.4)	80.3 (4.4)	<0.001
ADL	5.9 (0.4)	(3–6)			
IADL	7.7 (0.8)	(2–8)	7.8 (0.7)	7.2 (1.6)	<0.001
SPMSQ	9.6 (0.9)	(4–10)	9.7 (0.7)	9.2 (1.6)	<0.001
CIRS	1.95 (1.65)	(0–8)	1.9 (1.6)	2.2 (1.6)	0.226
MNA	26.97 (2.19)	(15.5–30)	27.1 (1.9)	25.8 (3.5)	<0.001
Exton-smith scale	19.38 (1.39)	(11–20)	19.5 (1.1)	18.5 (2.5)	<0.001
Medications	4.1 (2.7)	(0–13)	3.9 (2.6)	5.3 (3.0)	<0.001
Cohabitation status					0.895
living alone	152 (37.35)	-	133 (37.6)	19 (35.8)	
in an institution	1 (0.24)	-	1 (0.3)	0 (0.0)	
with family members	254 (62.61)	-	220 (62.1)	34 (64.2)	
Weight loss in the previous year	0.71 (3.18)	(−7.8–14.6)	0.8 (3.4)	0.2 (4.4)	0.227
Gait speed (m/sec)	1.08 (0.29)	(0.26–2.02)	1.1 (0.3)	0.9 (0.3)	<0.001
Hand grip strength test (Kg)	19.45 (10.63)	(2–55)	19.7 (10.6)	17.3 (11.0)	0.128
PASE	102.39 (56.02)	(2–398)	105.6 (55.0)	77.4 (58.4)	0.001
MPI score	0.23 (0.15)	(0.00–0.75)	0.21 (0.12)	0.34 (0.17)	<0.001
MPI categories					
MPI-1	330 (81.1)	-	304 (85.9)	26 (49.1)	<0.001
MPI-2	73 (17.9)	-	50 (14.1)	23 (43.4)	
MPI-3	4 (1.0)	-	0 (0.0)	5 (7.5)	
CHS index					0.002
Robust	261 (64.1)	-	235 (66.4)	26 (49.1)	
Pre-frail	109 (26.8)	-	94 (26.5)	15 (28.3)	
Frail	37 (9.1)	-	25 (7.1)	12 (22.6)	

Abbreviations: activities of daily living: ADL; instrumental ADL: IADL; Short Portable Mental State Questionnaire: SPMSQ; Cumulative Illness Rating Scale: CIRS; Mini Nutritional Assessment: MNA; Physical Activity Scale for the Elderly: PASE; Multidimensional Prognostic Index: MPI; Cardiovascular Health Study: CHS.

**Table 2 ijerph-18-12435-t002:** Comparison between Multidimensional Prognostic Index and Cardiovascular Health Study index in predicting overall mortality during five years of follow-up.

Item	C-Index	Standard Error	95% CI	*p* Value
MPI	0.69	0.04	0.61–0.77	<0.0001
CHS Index	0.59	0.04	0.52–0.67	<0.0001
Difference	0.10	0.04	0.02–0.18	0.01

Abbreviations: Multidimensional Prognostic Index: MPI; Cardiovascular Health Study: CHS; confidence intervals: CI.

## Data Availability

The data presented in this study are available on request from the corresponding author.

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
