# Peer review of "Multidimensional Frailty Predicts Mortality Better than Physical Frailty in Community-Dwelling Older People: A Five-Year Longitudinal Cohort Study"

_ijerph, 2021, doi:10.3390/ijerph182312435_

Round 1

Reviewer 1 Report

In this interesting and well written paper, Cella and colleagues show that multidimensional frailty, evaluated through the well recognized MPI tool, is superior to physical frailty in predicting 5-year mortality in a group of Italian community-dwellers. The findings are clear and relevant for clinical practice, since they contribute to define which tool is best for assessing frailty in geriatric medicine in terms of identification of the prognostic trajectory of each individual. 

I have the following comments: 

1) Physical frailty was measured in compliance with the Cardiovascular Health Study Criteria, that were modelled on the Fried frailty phenotype. However, the criteria used in the present investigation were not exactly the same as those used in the Cardiovascular Health Study, as stated by authors in lines 78-80. This circumstance should be discussed as a study limitation, especially for its potential of diminishing the predictive value of physical frailty on mortality.

2) Lines 129 onwards: it is unclear how the tripartite hierarchy was used for domain 7 (number of medications) and 8 (cohabitation status). Please specify. Furthermore, did the number of medications also consider topical drugs or not? 

3) Statistical analyses. Why was a ROC analysis not performed? The comparison of ROC curves of MPI and CHS in mortality prediction could be useful and add value to the study. 

4) The study population was mainly composed of fit older subjects (80.1% in the MPI-1 category and 64% in the robust category of CHS). Thus, the burden of frailty was indeed low. These characteristics may not completely coincide with those of older subjects attending geriatric outpatients' clinics or being hospitalized for acute complaints, which represent the intended target of use of the MPI tool. I suggest to discuss this circumstance as a limitation of the study. 

Author Response

1) Physical frailty was measured in compliance with the Cardiovascular Health Study Criteria, that were modelled on the Fried frailty phenotype. However, the criteria used in the present investigation were not exactly the same as those used in the Cardiovascular Health Study, as stated by authors in lines 78-80. This circumstance should be discussed as a study limitation, especially for its potential of diminishing the predictive value of physical frailty on mortality.

R: We fully agree with this comment. We have now added in the Limitations section the following statement:

“Finally, physical frailty was measured in compliance with the Cardiovascular Health Study Criteria, but not using the same criteria as those used in the Cardiovascular Health Study. Therefore, as widely known, this can decrease the predictive value of physical frailty in predicting mortality.[31]”

2) Lines 129 onwards: it is unclear how the tripartite hierarchy was used for domain 7 (number of medications) and 8 (cohabitation status). Please specify. Furthermore, did the number of medications also consider topical drugs or not? 

R: Thank you for the comment. The number of medications was categorized as <3, 4-7, >7 and not considering topical medications. As reported, cohabitation status included living alone, in an institution, or with family members.

3) Statistical analyses. Why was a ROC analysis not performed? The comparison of ROC curves of MPI and CHS in mortality prediction could be useful and add value to the study. 

R: Thanks for your comments. As stated in the Statistical Methods section, the discriminative capability of the two indexes was estimated in terms of Harrell’s C-index. This type of index is the equivalent of the AUC under the ROC curve used in the binary context. Harrell’s C and AUC belong to the same family of statistical indexes. The main difference between the AUC-ROC and the C-index is that the first assess the discriminative capability regardless the time in which the event onsets, the second is developed to assess the same thing but adjusting for the different time of events of each subject and for the presence of censoring.

4) The study population was mainly composed of fit older subjects (80.1% in the MPI-1 category and 64% in the robust category of CHS). Thus, the burden of frailty was indeed low. These characteristics may not completely coincide with those of older subjects attending geriatric outpatients' clinics or being hospitalized for acute complaints, which represent the intended target of use of the MPI tool. I suggest to discuss this circumstance as a limitation of the study. 

R: Thank you for this comment. Here we added a limitation in our Discussion section:

“In this regard, in fact, the study population was mainly composed of fit older subjects (80.1% in the MPI-1 category and 64% in the robust category of CHS), with an over-sampling of this population, being poorly generalizable in other contexts in which multi-dimensional and physical frailty have a high prevalence.”

Reviewer 2 Report

The authors assessed a large sample of elderly with CGA for frailty symptoms, including MPI and CHS, two models for multidimensional and physical frailty, respectively. After 5 years, they reported the mortality rate and found that MPI was more accurate in predicting it rather than CHS.

It is a remarkable longitudinal study, involving a large sample of participants, I have, though, some methodological questions:

  1. Introduction: please consider to include the following paper, since it involves multidimensional construct of frailty and a large international sample: Rainero, I., Summers, M. J., Monter, M., Bazzani, M., Giannouli, E., Aumayr, G., Burin, D., Provero, P., Vercelli, A. E., & My-AHA Consortium (2021). The My Active and Healthy Aging ICT platform prevents quality of life decline in older adults: a randomised controlled study. Age and ageing50(4), 1261–1267. https://doi.org/10.1093/ageing/afaa290
  2. The text in the introduction can be more readable, for example avoid the repetition of the word "frailty"
  3. Can you confirm that the MPI evaluation does not include any physical parameter? If yes, is that appropriate to call it "multidimensional" without the physical dimension?
  4. The categorization of MPI (low, moderate, high) requires a reference.
  5. Outcome mortality: you considered all mortality causes. Is that appropriate? How about deaths unrelated to health issues (e.g., car accidents)? I think you should be more specific and describe the causes of death.
  6. After the initial assessment, did you give the participants some feedback about their condition? if yes, they may have decided to start some intervention (physical or other) in order to improve their condition. I think this should be specified.
  7. did you not consider to do periodical follow-up about their conditions? why?
  8. please specify the onset between the assessments and the eventual deaths.
  9. more generally, please specify the descriptive statistics of those who are dead. I sincerely wonder if other factors (such as age) could predict even better than frailty the deaths.
  10. There are several typos and repetitions throughout the text. 

Author Response

Introduction: please consider to include the following paper, since it involves multidimensional construct of frailty and a large international sample: Rainero, I., Summers, M. J., Monter, M., Bazzani, M., Giannouli, E., Aumayr, G., Burin, D., Provero, P., Vercelli, A. E., & My-AHA Consortium (2021). The My Active and Healthy Aging ICT platform prevents quality of life decline in older adults: a randomised controlled study. Age and ageing, 50(4), 1261–1267. https://doi.org/10.1093/ageing/afaa290

R: We added this relevant reference, as suggested, in the Introduction section.

The text in the introduction can be more readable, for example avoid the repetition of the word "frailty"

R: Thank you so much for your comment. We have now removed the word frailty since redundant.

Can you confirm that the MPI evaluation does not include any physical parameter? If yes, is that appropriate to call it "multidimensional" without the physical dimension?

R: Here we confirm that MPI does not include any physical parameter, even if it is a multidimensional tool. In this sense, if you see the paper in which MPI was built, you can observe that several common tools used for predicting prognosis in older people were initially considered, including physical activity parameters, but the best set for predicting mortality was represented by the eight domains reported in the final version of the MPI.

The categorization of MPI (low, moderate, high) requires a reference.

R: Added.

Outcome mortality: you considered all mortality causes. Is that appropriate? How about deaths unrelated to health issues (e.g., car accidents)? I think you should be more specific and describe the causes of death.

R: Unfortunately specific causes of death are not available, but we know only the date and if the person was alive or not during follow-up. In our opinion, however, even if in younger persons it is true that overall mortality is often different from that related to health issues, in older people overall mortality is a good estimation of health-related mortality as also shown by our study and others indicating the use of prognostic tools in older people.  

After the initial assessment, did you give the participants some feedback about their condition? if yes, they may have decided to start some intervention (physical or other) in order to improve their condition. I think this should be specified.

R: Good point. After the initial assessment we did not recommend any specific intervention, also having in mind that is an observational study and that a consistent part of the people included was not frail.

did you not consider to do periodical follow-up about their conditions? why?

R: We did not plan to periodically evaluate these people, only verifying the survival status.

please specify the onset between the assessments and the eventual deaths.

R: Good point. Deaths were recorded until five years after the first assessment.

more generally, please specify the descriptive statistics of those who are dead. I sincerely wonder if other factors (such as age) could predict even better than frailty the deaths.

R: We sincerely thank the Reviewer for this comment. Accordingly, we have added a new Table 1 summarizing data by survival status. Moreover, we added a model including age and sex together that failed to show that these parameters are able to predict mortality.

There are several typos and repetitions throughout the text. 

R: We have carefully revised the paper for typos and repetitions, as indicated. Thank you for your careful reading.

Round 2

Reviewer 1 Report

The authors have adequately responded to all my previous comments. I have no further concerns.